# Rationale and protocol for a safety, tolerability and feasibility randomized, parallel arm, double-blind, placebo-controlled, pilot study of a novel ketone ester targeting frailty via immunometabolic geroscience mechanisms

**Brianna J. Stubbs**[1]*, **Gabriela Alvarez Azañedo**[1], **Sawyer Peralta**[1], **Stephanie Roa Diaz**[1], **Wyatt Gray**[1], **Laura Alexander**[1], **Wendie Silverman-Martin**[1], **Thelma Y. Garcia**[1], **Traci M. Blonquist**[2], **Vaibhav Upadhyay**[3,4], **Peter J. Turnbaugh**[3,5], **James B. Johnson**[6], **John C. Newman**[1,7]*

1 Buck Institute for Research on Aging, Novato, California, United States of America, 2 Biofortis, Mérieux NutriSciences, Addison, Illinois, United States of America, 3 Department of Microbiology & Immunology, UCSF, San Francisco, California, United States of America, 4 Independent Researcher, Greenbrae, California, United States of America, 5 Division of Geriatrics, UCSF, San Francisco, California, United States of America, 6 Department of Medicine, UCSF, San Francisco, California, United States of America, 7 Chan Zuckerberg Biohub-San Francisco, San Francisco, California, United States of America

* bstubbs@buckinstitute.org (BJS); jnewman@buckinstitute.org (JCN)

## Abstract

### Background

Frailty is a geriatric syndrome characterized by chronic inflammation and metabolic insufficiency that creates vulnerability to poor outcomes with aging. We hypothesize that interventions which target common underlying mechanism of aging could ameliorate frailty. Ketone bodies are metabolites produced during fasting or on a ketogenic diet that have pleiotropic effects on inflammatory and metabolic aging pathways in laboratory animal models. Ketone esters (KEs) are compounds that induce ketosis without dietary changes, but KEs have not been studied in an older adult population. Our long-term goal is to examine if KEs modulate aging biology mechanisms and clinical outcomes relevant to frailty in older adults.

### Objectives

The primary objective of this randomized, placebo-controlled, double-blinded, parallel-group, pilot trial is to determine tolerability of 12-weeks of KE ingestion in a broad population of older adults (≥ 65 years). Secondary outcomes include safety and acute blood ketone kinetics. Exploratory outcomes include physical function, cognitive function, quality of life, aging biomarkers and inflammatory measures.

### Methods

Community-dwelling adults who are independent in activities of daily living, with no unstable acute medical conditions (n = 30) will be recruited. The study intervention is a KE or a taste,

**Data Availability Statement:** No datasets were generated or analysed during the current study. All relevant data from this study will be made available upon study completion.

**Funding:** Funding for BIKE is provided by philanthropic donations from Dr James B. Johnson and from members of the Buck Institute Impact Circle. Dr Johnson assisted with conceptualization of the study and reviewed this manuscript but has no further role in study design, management, data collection, analysis, interpretation of data, decision to submit publications, or writing of publications. The Buck Institute Impact Circle has no role in conceptualization, study design, management, data collection, analysis, interpretation of data, decision to submit publications, review, or writing of publications. Dr Newman's participation in the study is supported by Buck Institute institutional funds. Dr Brianna Stubbs' participation in the study is supported by supported by the NIH (NIA) under award number K01AG078125.

**Competing interests:** The principal investigator (Dr. Newman), Dr. Brianna Stubbs, and the Buck Institute hold shares in BHB Therapeutics. Drs. Newman and Stubbs are inventors on patents relating to the use of ketone bodies that are assigned to The Buck Institute. All other authors have no conflicts to declare. Individual and institutional extensive conflict management plans were developed and approved by the Buck Institute and the IRB. Actions and decisions important to subject safety and study integrity are carried out by parties with no potential financial conflict. Participant consent is obtained by licensed registered nurses who have no financial conflict. Decisions on subject enrollment, continuation, and discontinuation are made by independent medical officers unaffiliated with Buck Institute and with no financial conflict. Data analysis for the primary outcome is carried out by an independent statistician with no financial conflict. Study staff, including the principal investigator, will maintain blinding through study completion unless unblinding is required for safety concerns. The principal investigator (Dr. Newman), Dr. Brianna Stubbs, and the Buck Institute hold shares in BHB Therapeutics. Drs. Newman (US 11,773,051 B2, US 11,608,306 B2) and Stubbs (US 11,645,228 B2) are inventors on patents relating to the use of ketone bodies that are assigned to The Buck Institute. This does not alter our adherence to PLOS ONE policies on sharing data and materials. All other authors have no conflicts to declare.

**Abbreviations:** 1-RM, 1 Repetition Maximum; ADL, Activities of Daily Living; AEs, Adverse Events; BDO, (R)-1,3-butanediol; BH-BD, Bis-

appearance, and calorie matched placebo beverage. Initially, acute 4-hour ketone kinetics after 12.5g or 25g of KE consumption will be assessed. After collection of baseline safety, functional, and biological measurements, subjects will randomly be allocated to consume KE 25g or placebo once daily for 12-weeks. Questionnaires will assess tolerability daily for 2-weeks, and then via phone interview at bi-monthly intervals. Safety assessments will be repeated at week 4. All measures will be repeated at week 12.

## Conclusion

This study will evaluate feasibility, tolerability, and safety of KE consumption in older adults and provide exploratory data across a range of aging-related endpoints. This data will inform design of larger trials to rigorously test KE effects on aging mechanisms and clinical outcomes relevant to frailty.

## Introduction

The geroscience approach of modulating fundamental aging mechanisms holds promise for generating new therapeutics for multifactorial conditions of aging which contribute to functional decline, disability, and loss of independence in older adults [1, 2]. Frailty syndrome is one such condition, conceptualized as a state of reduced physiological reserve and increased vulnerability to adverse health stresses that is a powerful risk factor for disability, institutionalization, and death [3]. Frailty is increasingly common with advancing age, with around 15% of adults over 65 years old in the United States meeting one standard definition [4]. The pathophysiology of frailty is complex, multisystem, and poorly understood, but is thought to include cellular energy production deficits along with chronic inflammation and immune dysfunction [5, 6]. These are among the molecular mechanisms often referred to as "Hallmarks of Aging", common mechanisms that change with age and underlie many diseases and conditions of aging [7]. Geroscience is the emerging field that seeks to manipulate these mechanisms to improve human health. The Geroscience Hypothesis, yet untested in human clinical trials, holds that intervening upon common underlying mechanisms of aging may ameliorate or prevent a variety of both specific chronic diseases of aging as well as multisystem geriatric syndromes such as frailty [1]. As of yet, there are no specific molecular therapies for frailty, although exercise programs, specialized geriatric care models, and nutritional interventions can positively alter the frailty phenotype and mitigate its consequences [8].

Nutrition and aging have long been intertwined through decades of study of the effects of dietary restriction and fasting on aging in animal models [9] and now humans (e.g. CALERIE [10], HALLO-P (NCT05424042)). A deepening understanding of the molecular mechanisms involved in fasting and dietary restriction has led to small molecule interventions now under investigation in clinical trials targeting various aging phenotypes, as well as a popular upswell of public and scientific interest in these dietary strategies. One key shared physiological element of such metabolic states is the production and utilization of ketone bodies. Ketone bodies, of which the most abundant is beta-hydroxybutyrate (βHB), are small molecules synthesized in the liver from lipids which circulate in the blood to provide non-glucose energy to extrahepatic tissues such as muscle, brain, heart, and other organs. Ketone bodies are synthesized and circulate constitutively, but are also strongly induced during fasting [11]. The energetic and molecular signaling activities of ketone bodies, including supporting

hexanoyl (R)-1,3-butanediol; BMI, Body Mass
Index; BO-BD, Bis-octanoyl (R)-1,3-butanediol;
BTQ, Beverage tolerability questionnaire; CRF, Case
Report Form; CSHA, Canadian Study of Health and
Aging; DSST, Digit Symbol Substitution Test; IADL,
Instrumental Activities of Daily Living; ITT,
Intention to treat; KEs, Ketone Esters; MoCA,
Montreal Cognitive Assessment; PLA, Placebo; PP,
Per protocol; QoL, Quality of Life; SASP,
Senescence-associated secretory phenotype;
SPPB, Short Physical Performance Battery; BHB,
Beta-hydroxybutyrate.

mitochondrial function and regulating inflammatory activation, support a mechanistic role in modulating aging and may be directly relevant to frailty [12]. We and others previously showed that a ketogenic diet, a dietary method of producing ketone bodies by reducing carbohydrate intake without fasting, extended healthy lifespan in wild-type mice including slowing overall aging-related functional declines, murine clinical frailty scores, and muscle strength declines [13, 14].

Ketone esters (KEs), such as bis-octanoyl-(R)1,3-butanediol, are small molecules that deliver ketone bodies without other dietary changes [15, 16]. KEs are hydrolyzed in the gut to release ketogenic fatty acids and a ketogenic alcohol, which are then metabolized in the liver to release ketone bodies [17]. In preclinical models, ketone bodies and KEs attenuate muscle atrophy through anticatabolic signaling activities [18, 19], improve heart function in age-related heart failure through direct energetic support [20, 21], and promote healthy function of T cell subpopulations [22, 23]. This hypothesis is further supported by demonstrated effects of KEs on blood glucose control [24–27], physical [28–30], cognitive [31–33], immune [34], and cardiovascular [33, 35] function in younger adult populations. These examples support our central hypothesis that ketone bodies delivered through KEs may ameliorate the frailty syndrome through multi-system energetic and signaling activities that improve metabolic and immune function. KEs have not been investigated in older adult populations.

Here we describe the protocol for a randomized, placebo-controlled, double-blinded, parallel-group pilot clinical trial with the objective of generating the first human safety and tolerance data for KEs in a generalizable population of older adults: The Buck Institute Ketone Ester (BIKE) Study. The placebo for BIKE contains canola oil, an inert, non-ketogenic fat, commonly found in the food supply. BIKE is designed as an example of a proof-of-concept geroscience clinical trial [2]: specifically assessing an intervention for tolerability and safety in a broad population of older adults, targeting specific aging-related molecular mechanisms, and linking those mechanisms to a clinically important outcome that is broadly representative of aging. We will collect exploratory data on clinical outcomes related to frailty and biological aging mechanisms, which will facilitate future planned work on clinical outcomes of KE consumption in a pre-frail or frail population.

## Methods

### Study design overview

The Buck Institute Ketone Ester (BIKE) Study is a randomized, double-blind, placebo-controlled, parallel arm, pilot study designed to assess the feasibility, safety, and tolerability of up to 25 g/day of KE ingestion for 12 weeks compared to a placebo in older adults ($\geq$ 65 years of age), and to generate exploratory data on endpoints related to aging and frailty (Figs 1 and 2). Following *Consent and Screening* (Visit 1), *Acute Ketone Kinetics* are measured during Visit 2, *Baseline* measures are collected at Visit 3 (week 0), daily KE or placebo consumption begins (day 1), an *Interim Safety* assesment is performed at Visit 4 (week 4), and *Final* post-intervention measures are collected at Visit 5 (week 12). Tolerance is assesed during phone calls every 2 weeks. The study protocol described here has been approved by Advarra Institutional Review Board (current version 1.2 approved on 17th Janurary 2023), any further ammendments will be reported to the sponsor and approved by the IRB. The study is registered with clincaltrials.gov (NCT05585762), all items from the WHO Trial Registration Dataset are included in S1 Table. Recruitment for the study began on November 28th 2023. Here we present details of the study protocol according to SPIRIT 2013 guidelines [36] (SPIRIT checklist provided in S1 Checklist).

| Visit Name | Phone Screen | Screening/ Consent Visit (1) | Acute Ketone Kinetics Visit (2) | Baseline Visit (3) | At Home | At Home | Interim Visit (4) | Phone Check In | Final Visit (5) |
|---|---|---|---|---|---|---|---|---|---|
| Time | | Day -56 to -21 | Day -42 to -7 | Day 0 | Days 1 to 14 | Days 15 to 84 | Week 4 | Weeks 2, 6, 8 and 10 | Week 12 |
| **SCREENING** | | | | | | | | | |
| Demographics | X | X | | | | | | | |
| Inclusion/Exclusion | X | X | X | X | | | X | X | X |
| Interview for Frailty Measures | X | X | | | | | | | X |
| Informed Consent | | X | | | | | | | |
| Review Medical History & Medication History | | X | | X | | | X | X | X |
| Vitals/Anthropometrics | | X | | X | | | X | | X |
| Blood and urine sampling – safety labs | | X | | X | | | X | | X |
| Randomization | | X | | | | | | | |
| **INTERVENTIONS** | | | | | | | | | |
| Acute ketone kinetics | | | X | | | | | | |
| Daily study product consumption | | | | | X | X | | | |
| **ASSESSMENTS** | | | | | | | | | |
| Capillary Blood Sampling | | | X | | | | | | |
| Assess tolerability | | | X | | X | X | X | X | X |
| Stool sample collection | | | X | X | X | | X | | X |
| Blood and urine archiving – biomarker analysis | | | | X | | | X | | X |
| Diet Recall | | | | X | | | | | X |
| Physical Testing | | | | X | | | | | X |
| Cognitive Testing | | | | X | | | | | X |
| Study Questionnaires | | | | X | | | | | X |
| Wrist actigraphy | | | | | X | X | | | |
| Continuous ketone monitor | | | | | X | | | | |
| Adherence monitoring | | | | | | | X | X | X |

**Fig 1. Schedule of BIKE study screening, interventions and assessments- SPIRIT Figure.**

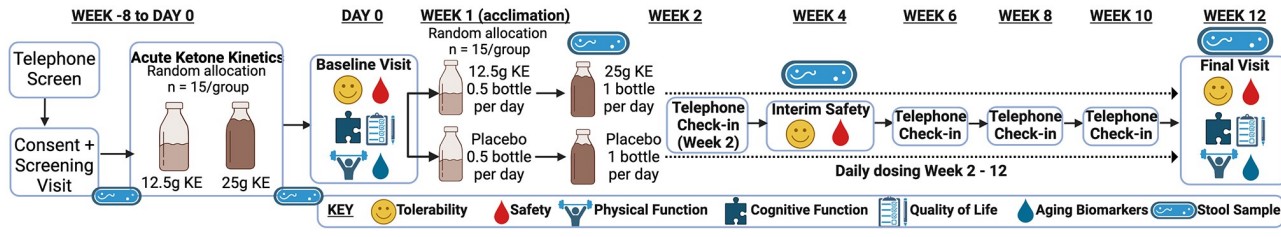

**Fig 2. BIKE study design schematic.**

## Study setting

The BIKE Study is conducted by investigators and clinical research staff at The Buck Institute for Research on Aging, an academic institutions based in Northern California, USA. Study procedures are conducted in the Clinical Research Unit at The Buck Institute. The majority of subjects are expected to be residents of Northern California.

## Study population

Subjects (n = 30) in the BIKE Study are community dwelling older adults ($\geq$ 65 years) in stable health, with roughly equal distribution of male and female subjects. Equal enrollment of males and females is important, as frailty is more common in women yet mortality is higher in men, and detailed pathophysiology may differ between sexes [37].

Inclusion criteria are broad, and exclusion criteria relatively limited, in order to enroll a broad sample of older adults, many of whom will have common chronic diseases and be taking multiple medications as in the intended future study population. Stable chronic disease and medications are not an exlcusion, except for specific items that may interact poorly with anticipated KE side effects such as gastrointestinal upset. Inclusion and exclusion criteria are found in Table 1:

## Study intervention and placebo

The BIKE Study KE beverage is a tropical-flavored beverage containing the KE bis-octanoyl (R)-1,3-butanediol (BO-BD) (BHB Therapeutics Ltd, Dublin, IRE). Each bottle is 75 mL and contains 25 g of KE. This KE was developed to resolve palatability issues with the first KE beverage launched by BHB Therapeutics (bis-hexanoyl (R)-1,3-butanediol, BH-BD). Both BO-BD and BH-BD are expected to be metabolized similarly, and deliver a rapid increase in circulating ketone concentrations [15, 16, 38, 39]. Subjects will consume the study beverage daily for 12 weeks at home within 5 minutes of breakfast. Half of a bottle (12.5 g KE) is consumed daily during week 1, and a full bottle daily (25 g KE) for the remainder of the study; this acclimation period is designed to reduce gastrointestinal issues and we have used it sucessfully in our studies of younger adults [40]. No modifications to this schedule are permitted, participants may discontinue at their request or at the discretion of the study medical officer. The placebo used in the study was custom manufactured by BHB Therapeutics. It is matched for volume, appearance, flavor, and calories; the KE is replaced by non-ketogenic canola oil. Subjects will be contacted by the study team every 2 weeks to review their study product use, which is expected to improve adherence and retention. Compliance will be assessed by daily completion of a paper Study Beverage Log reported by study subjects. The log will be compared to the number of unopened bottles returned at week 4 and week 12 visits. Subjects who consume between 80–120% of their allocated study product will be considered 'per protocol.'

**Table 1. Inclusion/exclusion criteria.**

| |
| --- |
| *Inclusion Criteria* |

- Men and women age $\geq$ 65 years
- BMI 18.5–34.9 kg/m$^2$ (inclusive at Screening/Consent Visit)
- No health conditions that would prevent them from fulfilling the study requirements as judged by the Clinical Investigator based on medical history and routine laboratory test results

| |
| --- |
| *Exclusion Criteria* |

- Non ambulatory
- Canadian Study of Health and Aging (CSHA) clinical frailty score > 5
- Required assistance with any activity of daily living except continence
- Living in an institutional setting
- Hospitalization within 30 days of the Screening/Consent Visit
- Abnormal laboratory test result(s) of clinical importance, indicating unstable chronic disease of major organ dysfunction (at the Screening/Consent Visit), at the discretion of the Clinical Investigator
- History or presence of uncontrolled and/or clinically active pulmonary, cardiac (e.g., > = New York Heart Association class III), hepatic, renal, endocrine (including type 1 diabetes), hematologic, immunologic, neurologic (e.g., Alzheimer's or Parkinson's diseases), psychiatric (including unstable depression and/or anxiety disorders) or biliary disorders. Stable chronic disease is not an exclusion criterion unless specified
- History or presence of a clinically important gastrointestinal condition that could potentially interfere with the evaluation of the study beverage, at the discretion of the Clinical Investigator, including inflammatory bowel disease, irritable bowel syndrome, chronic constipation, severe constipation, frequent diarrhea, surgery for weight loss, gastroparesis, systemic disease that might affect gut motility, reflux requiring daily medication, gastrointestinal ulcers or bleeding and clinically important lactose intolerance
- Uncontrolled hypertension (systolic blood pressure $\geq$140 mm Hg or diastolic blood pressure $\geq$90 mm Hg) as measured at the Screening/Consent Visit. One re-test will be allowed on a separate day prior to Visit 2
- Use of medications (over-the-counter or prescription) known to influence gastrointestinal function including, but not limited to, opioids, anti-diarrheal, and anti-spasmodic) within 30 days of the Screening/Consent, Acute Ketone Kinetics, or Baseline Visits
- Unstable use of thyroid, antihypertensive, antidepressant, or statin medications within 6 months of the Screening/Consent, Acute Ketone Kinetics, or Baseline Visits
- Undergoing treatment or active surveillance for cancer, or has been diagnosed with cancer in the prior two years (except for non-melanoma skin cancer)
- Recently used antibiotics within 30 days of the Screening/Consent, Acute Ketone Kinetics, or Baseline Visits
- Presence of a condition that would interfere with their ability to provide informed consent, comply with the study protocol, which might confound the interpretation of the study results, or put the subject at undue risk, at the discretion of the Clinical Investigator
- History of alcohol or substance abuse
- Consistent use of prescriptive or over-the-counter medications where alcohol is a contraindication, at the discretion of the Clinical Investigator
- Known allergy, intolerance, or sensitivity to any of the ingredients in the study beverages, including soy and milk protein
- Extreme dietary habits (e.g., intermittent fasting or time restricted eating, Atkins diet, vegan, very high protein/ low carbohydrate) within 30 days of the Screening/Consent, Acute Ketone Kinetics, or Baseline Visits
- Use of weight-loss medications (including over-the-counter medications and/or supplements) or programs within 30 days of the Screening/Enrollment, Acute Ketone Kinetics, or Baseline Visits
- Use of ketone supplements (ketone salts or esters, and medium chain triglyceride oils) within 30 days of the Screening/Consent, Acute Ketone Kinetics, or Baseline Visits
- Working at night or shifts that will impede maintaining a consistent meal schedule during the study
- Inability to confirm COVID-19 vaccination status
- (For female subjects) not have passed menopause, is pregnant, planning to be pregnant during the study period, lactating, or is of childbearing potential and is unwilling to commit to the use of a medically approved form of contraception throughout the study period

## Study outcomes

**Primary outcome.** The primary outcome of the BIKE Study is tolerability of the KE intervention, assessed using a beverage tolerability questionnaire (BTQ) [41, 42]. The BTQ includes ten symptoms: gas/flatulence, nausea, vomiting, abdominal cramping, stomach rumbling, burping, reflux (heartburn), diarrhea, headache, and dizziness. Subjecs are either asked if the issue occurred since they took the study beverage (paper questionnaire completed daily 3–6 h after KE consumption during 2-week acclimation period) or during the last week (completed verbally during bi-weekly telephone call), and asked to rank the intensity as none, mild, moderate or severe. These correspond to scores of 0–3 respectively for each issue, giving a maximal composite score of 30. After acclimation, subjects are asked to report the number of days that each symptom occurred.

The key tolerability outcome is the proportion of subjects reporting the same moderate to severe symptom (among dizziness, headache, or nausea) occurring on more than one day within any given two week recall period (after week 0–2 acclimation period).

Headache, nausea, and dizziness were selected as they occurred significantly more often at mild severity in our previous 28-day study of a KE in healthy adults [40]. The 'moderate to severe' intensity was selected as 'mild' symptoms are common in this population [43], and may be acceptable if they are outweighed by benefits of the intervention. Moderate to severe symptoms are less common [44] and more likely to be deterimental to quality of life or function. The frequency of 'more than one day per period' is used to reduce the impact of idiosyncratic symptoms on the analysis given the common occurance of such symptoms in the target population.

**Secondary outcomes.** Safety is a secondary outcome of the BIKE Study; assessed through changes over time in clinical and laboratory measures throughout the study within each group. Laboratory tests run at screening and blood samples at *Baseline*, *Interim Safety* and *Final Visits* (weeks 0, 4, and 12) include hematology, chemistries, liver function tests, and thyroid function tests. Based on previous data on KE effects, we are particularly focused on potassium [45, 46] and bicarbonate [46, 47] along with liver function tests as KEs exploit natural hepatic metabolism in generating ketone bodies. Changes in body weight, heart rate, systolic and diastolic blood pressure, and orthostatic changes in blood pressure will be followed at the last three study visits. KEs may increase resting heart rate [45, 48] and increase blood flow to the brain and heart via vasodilation [45, 48, 49] which, while potentially therapeutic, require monitoring. KEs are not associated with weight loss [40, 50, 51], but body weight is followed to ensure that gastrointestinal side effects or taste do not cause clinically meaningful weight loss. The key safety outcome is the longitudinal within-group change in safety blood parameters. Our previous 28-day study of healthy adults did not find any clinically meaningful changes in labs [40], nor were any present in other studies of KEs [50, 51]. However, our target population will likely include a greater proportion of potentially interacting comorbidities and medications compared to these prior studies.

A further secondary outcome is monitoring the acute changes in blood ketone and glucose conentrations for 4 h post-consumption of a serving of KE, which has not be studied in an exclusively older adult cohort. At the *Acute Ketone Kinetics* Visit (Visit 2), subjects will be randomly allocated to consume either 12.5g or 25 g of KE after a standard breakfast. Prior to KE ingestion, and at regular intervals for 4 h following, capillary blood samples will be obtained from a finger and analysed instantly for BHB and glucose concentrations using a handheld device (Keto Mojo, CA, USA). We will compare the changes over the Acute Ketone Kinetics study day both from t = 0 and between the serving size groups. A subset of subjects will wear a continous ketone monitor (Abbott, IL, USA) for the first two weeks of the study, to provide data confiming the repeatability of BHB kinetics at home.

**Exploratory outcomes.** Exploratory outcomes include longditudinal changes in physical and cogntitive function, quality of life (QoL), and geroscience biomarkers in the KE and placebo groups. These are assesed at *Baseline* (Visit 3, week 0) and *Final*, post-intervention (Visit 5, week 12) timepoints unless otherwise stated; notably no study beverage is consumed on the morning of any study visits.

Exploratory physical function measures will be collected at week 0 and 12 as follows: 1 repetition maximum (1-RM) leg press strength, 70% 1-RM leg press fatiguability, 4 m gait speed, Short Physical Performance Battery (SPPB), 4 m gait speed is a component of SPPB but analyzed independently as well) [52], 6-minute walk test [53], grip strength via dynamometer, and actigraphy via FitBit. A recent consensus statement from the International Conference on Frailty and Sarcopenia Research Task Force on designing drug trials in frailty recommends these measures [54]. Similar outcomes were used in the CRATUS trial of mesenchymal stem cells [55], ENRGISE trial of losartan and fish oil [56], LIFE physical activity trial [57], and a senolytic clinical trial [58], and are proposed for TAME [59]. The selected measures have strong predictive value for disability [8, 60, 61]. Physical performance is also likely to be more dynamic in a 12-week study than longer-term clinical measures such as chronic diseases or disability.

The exploratory cognitive measures assesed at week 0 and 12 include a standard cognitive screen (Montreal Cognitive Assessment or MoCA) [62], as well as quantitative measures of processing speed which are more likely to improve with the intervention within 12 weeks. (Trails A, Trails B [63], and the Digit-Symbol Substitution Test or DSST [64]). MoCA and Trails A/B are components of the National Alzheimer's Coordinating Center Uniform Data Set Version 3 [65].

Other exploratory geriatric outcome measured at week 0 and 12 include Activities of Daily Living (ADLs), Instrumental Activities of Daily Living (IADLs), Fried Frailty Index, and CSHA Clinical Frailty Score, which are crucial assessments of overall health although unlikely to change in 12 weeks [66]. Quality of life measures were chosen based on relevance to frailty and factors that have been anecdotally reported to change with KE use. QoL will be assessd using: SF-36 [67], Profile of Mood States–Short Form [68], Sexual QoL Questionnaire (Male or Female version), Pittsburgh Sleep Quality Index [69], Pittsburgh Fatiguability Scale [70], and Geriatric Depression Scale [71].

Biomarkers in geroscience interventional clinical trials may include those related to the pharmacokinetics or downstream mechanisms of the intervention, and biomarkers [2] to help test if the intervention broadly impacts aging as per the Geroscience Hypothesis [72]. We will test exploratory biomarkers at week 0 and week 12 including: 1) blood tests identified by the TAME Biomarkers Workgroup as potentially useful in an interventional trial [72], 2) composite biomarkers using routine laboratory tests, 3) comprehensive 69-mer cytokine panel (via the TGN Facility for Geroscience Analysis), 4) DNA methylation for use in aging clock models and 5) alterations in gut microbiome using stool sample analysis at intervals through the study.

The TAME Biomarkers Workgroup formulated criteria for biomarkers in geroscience clinical trials that include feasibility and reliability, biological plausibility, robust association with risk, and responsiveness to interventions [72]. This resulted in a list of 9 biomarkers: IL-6, CRP, sTNFR1, Cystatin C, IGF-1, NT-proBNP, insulin, GDF15, and IGFBPs. IL-6, CRP, and sTNFR1 are implicated in frailty [73–75], and along with insulin, IGF-1, and IGFBPs, may be directly or indirectly regulated by ketone bodies. The comprehensive 69-mer cytokine panel is optimized for immunosenescence and SASP and includes TNFα, IFNγ, IL-6, IL-8, IL-10, IL-17, IL1β, sTNFR1/2, and more. We will apply three composite aging biomarkers derived from routine laboratory tests: Belsky, FI-LAB and DNA methylation. Belsky et al. developed two

biomarkers based on the Klemera-Doubal Method and a homeostatic dysregulation model and showed that the dietary restriction intervention in CALERIE slowed both measures [76]. FI-LAB is a clinical frailty index derived from 21 common laboratory tests plus blood pressure that predicts mortality among older adults [77, 78]. We will isolate peripheral blood mononuclear cells for DNA methylation analysis which will allow application of aging 'clocks.'

The effect of KE on the human microbiome is not yet known. Ketogenic diet rapidly alters the microbiome in both humans and mice, and in mice both KD and KE can reduce systemic pro-inflammatory Th17 cells and alter the gut microbiota [22]. To address this we will collect five stool specimens from each subject (before Visit 2, before Visit 3, one week post-Visit 3, immediately post- week 4, at week 12) to identify acute and chronic (across the 20 weeks) changes in the microbiome.

## Subject timeline

For all in-person visits, subjects must meet the following pre-test criteria: fasting ≥10h, no alcohol ≥10h, no exercise ≥10h, no pyschoactive cannabis products ≥10h. They will be queried on arrival to confirm compliance. At every visit, subjects will be queried on their use of any new medications and supplement, or any changes to their health. Subjects will also be queried to ensure compliance with study protocol, and will be reminded of study instructions. A summary of procedures is shown in Fig 1.

**Telephone screening.** Subjects will complete a telephone screening to assess initial eligibility by confirming their age, that they are independent on elements of activities of daily living (ADL), any allergies, and the absence of recent hospitalizations.

**Screening visit (Visit 1).** Subjects will provide written informed consent after discussing the study procedures with a registered nurse and prior to collection of blood and urine samples, demographic information, medical history, ADL, IADL and CSHA frailty scoring, and measurement of vital signs by a registered nurse. These data will be reviewed by an independent study medical officer who will make final decisions on eligibility and enrollment into the study. Subjects will be asked to taste the study beverage and confirm that they are willing to consume it daily for 12-weeks. Subjects will receive a stool sample kit for at-home collection prior to Visit 2.

**Acute ketone kinetics visit (Visit 2).** Subjects will be randomly allocated to either 12.5g or 25 g of KE following a standard breakfast (oatmeal with protein powder). Before and 4 h after study beverage consumption, subjects will verbally complete the BTQ described above, to determine the incidence of any side-effects during the visit. Capillary blood samples will be obtained by a research associate from a finger using a lancing device and analyzed instantly for BHB and glucose concentrations using a handheld device (KetoMojo, CA, USA). Collection time points are: pre-consumption, 30-, 60-, 90-, 120-, 180- and 240-minutes post-consumption. Study staff are not blinded to serving size for this visit. Subjects will be familiarized with the physical function tests that will be completed at Visit 3, and receive instruction on how to capture one day's food intake with an online diet log (ASA-24, NCI [79]) and be given a FitBit Inspire 2 device (FitBit, CA, USA) to wear for the remainder of the study in addition to a second stool sample kit for at-home collection prior to Visit 3.

**Baseline visit (Visit 3, week 0).** Fasted blood and urine samples will be collected for analysis as described above. Vital signs will be assessed after subjects have completed QoL questionnaires. Physical and cognitive function will be tested as described above; tests will be performed by a trained and certified (where appropriate) registered nurses or research associates. A continuous ketone monitor will be applied to a subset of subjects. Subjects will be given the Study Beverage Log that will be completed each day at home, a stool sample kit for at-

home collection one week after Visit 3, and a one-month supply of their study beverage, plus overage (35 bottles).

**At-home procedures.** Subjects will consume the study beverage every day for 12-weeks within 5 minutes after their breakfast. For the first week, they will consume half of a bottle daily (12.5 g KE), and for the remainder of the study they will consume a full bottle daily (25 g KE). They will complete a Study Beverage Log to indicate time of product consumption. Additionally, for the first two weeks (acclimation period), a full BTQ will be completed each day. For the remainder of the study, an open-ended question will allow subjects to record any side effects that they experience each day. The Study Log will be reviewed for compliance and BTQ outcomes during bi-weekly phone calls (week 2, 6, 8 and 10) and at Visit 4 and Visit 5. Adverse events (AEs) will also be queried during each phone call. The final month's supply of product will be mailed to subjects during week 7, along with a stool sample kit for at home collection up to 3 days before Visit 5.

**Interim safety visit (Visit 4, week 4).** Tolerability will be assessed by verbal completion of the BTQ, and AEs will be queried. Fasted blood and urine samples will be collected for safety lab analysis. Adherence will be monitored by review of the Study Beverage Log and counting of unused bottles of product brought in by the subject. One months' supply of product plus overage (35 bottles) will be provided for subjects to take home, along with a new Study Beverage Log. Subjects will be given a stool sample kit for at home collection 3 days after Visit 4. The independent study medical officer will review the safety lab results to approve continuation of subjects in the study.

## Final visit (Visit 5, week 12)

The Final Visit is identical to the Baseline Visit, except that the following will not take place: application of continuous ketone monitor, provision of study beverage, and Study Beverage Log distribution. The following will take place: collection of Study Beverage Log, compliance monitoring, verbal BTQ administration, and AEs query.

## Sample size

Based on studies of symptoms in older adults [43, 44], we predict the number of subjects who experience more than 2 days of moderate or severe headache, nausea or dizziness in a 2-week period (primary outcome) rate in the placebo group will be approximately 10% of the group. N = 30 (15 per arm) was selected as it was a feasible size for this pilot study that provides 36% power to detect a meaningful 25% increase (from 10% to 35%) in the proportion of subjects meeting our primary outcome in the KE condition (i.e., the KE is poorly tolerated) with two-sided $\alpha = 0.10$. If we assume a larger deleterious effect of the KE intervention, n = 30 (15 per arm) provides approximately 75% power to detect a 45% increase (from 10% to 55%) in the proportion of subjects meeting this primary outcome in the KE condition, with two-sided $\alpha = 0.10$.

## Recruitment

Initial recruitment is via sharing of study information on the Buck Institute website, outreach to a voluntary mailing list of interested individuals maintained by the Buck Institute, and via response to public outreach including a local newspaper article and talks at community group events. Further recruitment plans include ongoing engagement with community groups serving older adults, outreach to organzations representing communities underrepresented in medicine, and flyer mailings targeted to communities underrepresented in medicine.

## Allocation and blinding

Study beverage sequence allocation will occur randomly, based on a block allocation sequence (block size 4) prepared by a statistician. It will result in a 1:1:1:1 breakdown of subjects to receive either 12.5 or 25 g of KE at the *Acute Ketone Kinetics* Visit, and then either KE or placebo beverage for the 12-week study. Stratification by sex will occur to ensure equal allocation to each group. No other stratification will occur. Allocation will occur at Visit 2, with the sequence assigned by the research team in the order of subject enrollment.

Study beverage will be coded and labelled by external personnel and all study staff will remain blinded to the code allocation of the beverage throughout the study, but not the group allocation of the subject. Unblinding will occur after compeletion of all subjects and after analysis of the primary data outcome by an external statistician. Written procedures govern emergency unblinding of individuals or the study as a whole. Individual subjects are provided with a pocket card including their allocation code and a description of anticipated possible clinical effects of KE. They will have their allocation revealed to them and their health care providers via the study medical officers (in consultation with the principal investigator) if it is determined that unblinding is necessary for the care of an adverse event. In such an instance, allocation coding will remain blinded to the principal investigator, study staff, and all other subjects.

## Subject retention

Retention will be facilitated by building rapport with subjects during early study visits and ongoing bi-weekly contact with study subjects via telephone calls as well as email reminders of upcoming visits. In addition, stipends will be paid at the end of each visit; the value of the stipend increases each visit to incentivize continued participation.

## Data management and dissemination

Data will be entered into paper case report forms (CRFs) during study visits by research associates or registered nurses and reviewed for completeness by an investigator before it is transcribed into an electronic datasheet. Every effort will be made to maintain subject confidenitality before, during, and after the study. CRFs and entries into the electronic data sheet are idenitifed by subject number, and the code-break linking the subjects' name to their study number is stored seperately to the CRFs. CRFs are kept in secure cabinet in a room with limited access. The electronic datasheet is stored on a secure internal server at The Buck Institute. Data will be spot-checked for quality during the study closure, and retained for up to 5 years after closure.

Research data will be shared according to the most recent NIH guidelines. We are committed to the sharing of final research data, being mindful that the rights and privacy of people who participate in research must be protected at all times and that there is the need to protect patentable and other proprietary data. Any datasets will be free of any identifiers that would permit linkages to individual research participants and variables that could lead to deductive disclosure of individual subjects. Anonymized raw, subject-level data will be held at The Buck Institute under the custody of the Principal Investigator and de-identified data will be made available upon reasonable written request to the PI. The results of the study will be disseminatied via publication in scientific papers, abstracts, and presentations. Authorship will be determined by ICMJE guidelines, manuscripts will be prepared by the study team.

## Statistical analysis

All analyses will be based on intention to treat and will be conducted by an external study statistician who will be blinded to intervention coding. The proportion of subjects with at least 2 occurrences of the target symptoms during the post acclimation period will be estimated along with a 90% confidence interval within each group. For the primary outcome, the CI will be estimated with the exact binomial confidence interval. The proportion of subjects with the primary outcome will be compared between groups with Fisher's exact test. The frequency of each individual symptom and the frequency of mild symptoms will be explored with categorical data analysis methods which includes confidence interval estimation. The total composite scores will be evaluated with a Wilcoxon rank sum test. During the 14-day acclimation period, the daily composite score will be calculated as the sum of the individual items and will be analyzed with a random coefficient model. The within-group change in safety labs will be compared between groups with a Wilcoxon rank sum test and a false discovery rate adjustment to control for multiple comparisons. Changes in continuous secondary and exploratory outcomes will be compared between groups with an analysis of covariance approach or a repeated measures model, confidence intervals will be estimated as for the primary outcome or based on the t-distribution as appropriate. The intent-to-treat (ITT) population will comprise data for all subjects who were randomized and consumed one serving of the study beverage. In addition, the per protocol (PP) population will be identified as a subset of the ITT population, in which subjects will be excluded for the following reasons and possibly others: violations of inclusion or exclusion criteria that could influence the evaluation of response, non-compliance by the subject, including, but not limited to: use of prohibited drugs or any products thought to alter the primary outcome variable during the study, less than 80% or more than 120% compliance with study beverage consumption, or not adhering to instructions as outlined in the protocol. All decisions regarding exclusion from the ITT and PP populations will be documented prior to database lock. Missing data will not be imputed.

## Safety monitoring and oversight

Remote and on-site monitoring visits will be conducted periodically during the study, focusing on human subject protection and data integrity risks of the trial. A clinical monitoring plan will be developed which identifies specific risk-based monitoring focal points. These may include:, informed consent, eligibility criteria, adverse events, protocol deviations, endpoints, study beverage accountability.

Independent study medical officers will be retained tomonitor the study data, they will be practicing physicians with experise in geriatric medicine. The medical officers will make final determinations on all study enrollment, continuation (*Interim Safety Visit*), and discontinuation decisions, as well as review any serious adverse events and events of note, and consult with the study team on any other adverse events. The Buck Institute designated an institutional officer to represent the Instiute and the philanthropic funders.

Adverse events will be queried at visits and telephone calls and recorded in the CRF. Serious adverse events will be reported to the medical officers, institutional officer, and the IRB. 'Events of note' are a subset of adverse events that result in discontinuation from the study, require treatment or require diagnostic evaluation. No interim analysis is planned, however if a pre-determined threshold of excess events of note occuring in one of the study groups is reached, or if a serious adverse event occurs the committee comprising of the medical officers, institutional officers, and principal investigator will convene to discuss unblinding and study pausing or discontinuation. No medical care will be provded during the study and no form of compensation is offered for any injury.

## Discussion

This pilot study will provide important new information on the application of ketone body biology to human health and disease in aging. Firstly, KEs have not been rigorously tested in older adults, therefore the data resulting from this study will support further research with KEs in other age-related conditions. Secondly, the particular KE under investigation here is more palatable than existing products, and as it requires endogenous production of ketone bodies from the fatty acid components it more closely mimics dietary ketosis. Thirdly, this will be the most direct test to date of KE effects on geroscience pathways in humans, beginning to extend into humans the extensive pre-clinical literature linking ketosis to aging mechanisms. Fourthly, as ketone bodies are generated endogenously during both caloric restriction and exercise, KE can provide more specific mechanistic insight into the clinical effects of such interventions as in the NIH-funded CALERIE and HALLO-P trials. Finally, this is intended to be an example of an early proof-of-concept clinical trial testing the biological effects of a geroscience-guided intervention, with many elements of its design based on frameworks developed by the NIH-funded Translational Geroscience Network [2]. These design and conceptual elements include the goal of specifically assessing safety in an older population, focusing on a broadly representative and generalizable population, the use of functional outcome measures, and the combination of intervention-specific with general aging biomarkers. Translational geroscience is an emerging field, and early-stage clinical trials have been identified as the key bottleneck in the development of interventions, as well as being critical for training the next generation of investigators [80].

The need to carefully balance safety in a less resilient population with the desire to generate meaningful data in our exploratory outcomes led to the selection of the 12-week study duration. Studies of KEs in younger adults have demonstrated safety over a 28-day study period, hence 12-weeks represents a significant extension of the existing data. The Interim Safety Visit (Visit 4, week 4) is designed to ensure that subjects do not continue in the study if clinically meaningful negative changes to blood safety labs occur. Assuming positive safety and tolerability findings in this population, the study duration could be extended in future work.

Similarly, serving size selection for this study took into consideration multiple factors including feasibility, safety, and likely efficacy. Some functional endpoints, such as cognition [81] and cardiac function [45], appear to be responsive to the blood concentration of BHB, hence giving the maximal tolerable KE serving size could increase the chance of observing KE efficacy. Whilst some KEs have been studied at serving sizes up to 75 g/day for 28 days, the KE family used in this study has only been studied up to 25 g/day for 28 days, hence why the 25 g serving size was selected for this first-in-older-adults 12-week study. Splitting the daily serving into multiple smaller boluses through the day may have provided greater total time in nutritional ketosis for each study day, however there is no definitive evidence to support increased time in ketosis vs. a higher BHB peak. Increasing the subject burden and thus increasing the chance of non-adherence or drop-outs by requiring twice daily consumption was identified as a more important consideration than consistent ketosis coverage, hence why 25 g/day servings in one bolus was chosen. If safety data for this KE were expanded, larger serving sizes could be implemented in future work.

Study population selection was another area of careful consideration. Whilst the study is intended to recruit healthy individuals, older adults unavoidably have a greater incidence of concurrent health conditions and medication use. Requiring no medication use and no active conditions would be prohibitive for recruitment in an older adult population and would both exclude the pathophysiology of interest while sharply limiting relevance to the majority of older adults. With this in mind, we set intentionally broad inclusion and exclusion criteria; as

subjects must be independent for all activities of daily living (excluding self-managed continence issues) and we explicitly exclude a range of relevant active and unstable conditions, this will exclude most actively unwell individuals. The aim is to include healthy older adults, accepting that this includes individuals who are managing some health conditions.

The field of geroscience translational clinical trials is relatively new and standardization of outcome measures across studies presents both challenges and opportunities [2]. The outcome measures included in this study were chosen based on their hypothesized sensitivity to our KE intervention and to harmonize between other innovative geroscience clinical trials, the latter representing a major strength of this protocol. The ability to compare between study populations will inform the clinical significance of any changes following the KE intervention compared with other proposed geroscience interventions. Although some of the exploratory outcome measures are unlikely to change over the study period in a healthy population (i.e., ADL, IADL, SPBB, MoCA), others may be more dynamic (i.e., blood biomarkers, grip strength, 1-RM). Future work in a frail or pre-frail population, with a longer study period and increased KE serving size, would increase the probability of detecting significant changes in these measures.

## Sources of funding, author declarations and conflict management

Funding for BIKE is provided by philanthropic donations from Dr James B. Johnson and from members of the Buck Institute Impact Circle. Dr Johnson assisted with conceptualization of the study and reviewed this manuscript but has no further role in study design, management, data collection, analysis, interpretation of data, decision to submit publications, or writing of publications. The Buck Institute Impact Circle has no role in conceptualization, study design, management, data collection, analysis, interpretation of data, decision to submit publications, review, or writing of publications.

Dr Newman's participation in the study is supported by Buck Institute institutional funds. Dr Brianna Stubbs' participation in the study is supported by supported by the NIH (NIA) under award number K01AG078125.

The KE intervention is provided gratis by BHB Therapeutics Ltd (Ireland). BHB Therapeutics also arranged for manufacture of the matched placebo, paid from BIKE study funds. BHB Therapeutics markets formulated KE beverages to consumers. BHB Therapeutics provided no funding for the study, and has no role in the design, management, data collection, analysis, interpretation of data, decision to submit publications, or writing of publications.

The principal investigator (Dr. Newman), Dr. Brianna Stubbs, and the Buck Institute hold shares in BHB Therapeutics. Drs. Newman (US 11,773,051 B2, US 11,608,306 B2) and Stubbs (US 11,645,228 B2) are inventors on patents relating to the use of ketone bodies that are assigned to The Buck Institute. This does not alter our adherence to PLOS ONE policies on sharing data and materials. All other authors have no conflicts to declare. Individual and institutional extensive conflict management plans were developed and approved by the Buck Institute and the IRB. Actions and decisions important to subject safety and study integrity are carried out by parties with no potential financial conflict. Participant consent is obtained by licensed registered nurses who have no financial conflict. Decisions on subject enrollment, continuation, and discontinuation are made by independent medical officers unaffiliated with Buck Institute and with no financial conflict. Data analysis for the primary outcome is carried out by an independent statistician with no financial conflict. Study staff, including the principal investigator, will maintain blinding through study completion unless unblinding is required for safety concerns.

## Supporting information

**S1 File.**
(DOCX)

**S1 Table.**
(DOCX)

**S1 Checklist. SPIRIT 2013 checklist: Recommended items to address in a clinical trial protocol and related documents\*.**
(DOC)

## Author Contributions

**Conceptualization:** John C. Newman.

**Formal analysis:** Traci M. Blonquist.

**Funding acquisition:** Brianna J. Stubbs, John C. Newman.

**Investigation:** Brianna J. Stubbs, Sawyer Peralta, Stephanie Roa Diaz, Laura Alexander, Wendie Silverman-Martin, Vaibhav Upadhyay, Peter J. Turnbaugh.

**Methodology:** Brianna J. Stubbs, Gabriela Alvarez Azañedo, Sawyer Peralta, Stephanie Roa Diaz, Wyatt Gray, Traci M. Blonquist, Vaibhav Upadhyay, Peter J. Turnbaugh, James B. Johnson, John C. Newman.

**Project administration:** Brianna J. Stubbs, Thelma Y. Garcia, John C. Newman.

**Resources:** Brianna J. Stubbs.

**Supervision:** Brianna J. Stubbs, Thelma Y. Garcia, John C. Newman.

**Writing – original draft:** Brianna J. Stubbs, Gabriela Alvarez Azañedo.

**Writing – review & editing:** Brianna J. Stubbs, Sawyer Peralta, Stephanie Roa Diaz, Wyatt Gray, Laura Alexander, Wendie Silverman-Martin, Thelma Y. Garcia, Traci M. Blonquist, Vaibhav Upadhyay, Peter J. Turnbaugh, James B. Johnson, John C. Newman.

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
