## [Decision Letter · Decision Letter 0]

1 Dec 2023

PONE-D-23-22898Rationale and protocol for a safety, tolerability and feasibility randomized, parallel group, double-blind, placebo-controlled, pilot study of a novel ketone ester targeting frailty via immunometabolic geroscience mechanisms.PLOS ONE

Dear Dr. Stubbs,

Thank you for submitting your manuscript to PLOS ONE. After careful consideration, we feel that it has merit but does not fully meet PLOS ONE’s publication criteria as it currently stands. Therefore, we invite you to submit a revised version of the manuscript that addresses the points raised during the review process.

We look forward to receiving your revised manuscript.

Kind regards,

Ian James Martins, PhD

Academic Editor

PLOS ONE

2. Our internal editors have looked over your manuscript and determined that it is within the scope of our Aging in Human Health and Disease Call for Papers. This call for papers aims to highlight the excellent work being done by researchers across the world on the subject of aging. Additional information can be found on our announcement page: https://collections.plos.org/call-for-papers/aging-in-human-health-and-disease/. If accepted, your submission will be included within the collection. Please note that being considered for the Collection does not require an additional peer review beyond the journal’s standard process and will not delay the publication of your manuscript if it is accepted by PLOS ONE. If you have any questions or concerns about this process, please contact the journal at plosone@plos.org

3. We note that you have a patent relating to material pertinent to this article. Please provide an amended statement of Competing Interests to declare this patent (with details including name and number), along with any other relevant declarations relating to employment, consultancy, patents, products in development or modified products etc. Please confirm that this does not alter your adherence to all PLOS ONE policies on sharing data and materials, as detailed online in our guide for authors http://journals.plos.org/plosone/s/competing-interests by including the following statement: "This does not alter our adherence to  PLOS ONE policies on sharing data and materials.” If there are restrictions on sharing of data and/or materials, please state these. Please note that we cannot proceed with consideration of your article until this information has been declared.

5. We note that the original protocol file you uploaded contains a confidentiality notice indicating that the protocol may not be shared publicly or be published. Please note, however, that the PLOS Editorial Policy requires that the original protocol be published alongside your manuscript in the event of acceptance. Please note that should your paper be accepted, all content including the protocol will be published under the Creative Commons Attribution (CC BY) 4.0 license, which means that it will be freely available online, and any third party is permitted to access, download, copy, distribute, and use these materials in any way, even commercially, with proper attribution.

Therefore, we ask that you please seek permission from the study sponsor or body imposing the restriction on sharing this document to publish this protocol under CC BY 4.0 if your work is accepted. We kindly ask that you upload a formal statement signed by an institutional representative clarifying whether you will be able to comply with this policy. Additionally, please upload a clean copy of the protocol with the confidentiality notice (and any copyrighted institutional logos or signatures) removed.

6.Please review your reference list to ensure that it is complete and correct. If you have cited papers that have been retracted, please include the rationale for doing so in the manuscript text, or remove these references and replace them with relevant current references. Any changes to the reference list should be mentioned in the rebuttal letter that accompanies your revised manuscript. If you need to cite a retracted article, indicate the article’s retracted status in the References list and also include a citation and full reference for the retraction notice.

Additional Editor Comments:

The authors need to ensure that the research is properly verified and need to improve the quality of the manuscript. The authors need to hone in on the key points raised by the three reveiwers and correct inadvertent errors in the revised manuscript.

Reviewers' comments:

Reviewer's Responses to Questions

**Comments to the Author**

1. Does the manuscript provide a valid rationale for the proposed study, with clearly identified and justified research questions?

Reviewer #1: Yes

Reviewer #2: Yes

Reviewer #3: Yes

2. Is the protocol technically sound and planned in a manner that will lead to a meaningful outcome and allow testing the stated hypotheses?

Reviewer #1: Yes

Reviewer #2: Yes

Reviewer #3: Yes

3. Is the methodology feasible and described in sufficient detail to allow the work to be replicable?

Reviewer #1: Yes

Reviewer #2: Yes

Reviewer #3: Yes

4. Have the authors described where all data underlying the findings will be made available when the study is complete?

Reviewer #1: Yes

Reviewer #2: Yes

Reviewer #3: Yes

5. Is the manuscript presented in an intelligible fashion and written in standard English?

Reviewer #1: Yes

Reviewer #2: Yes

Reviewer #3: Yes

6. Review Comments to the Author

You may also provide optional suggestions and comments to authors that they might find helpful in planning their study.

Reviewer #1: The primary objective of this randomized, placebo-controlled, double-blind, parallel group, pilot trial is to determine tolerability of 12-weeks of KE ingestion in a generalizable population of older adults (≥ 65 years). Secondary outcomes include safety and acute blood ketone kinetics. Exploratory outcomes include physical function, cognitive function, quality of life, aging biomarkers, and inflammatory measures. Overall, this study will evaluate feasibility, tolerability, and safety of KE consumption in older adults and provide exploratory data across a range of geroscience-related endpoints.

All key components of the trial protocol were discussed well, including the experimental design, primary objective, primary endpoint, secondary endpoints, exploratory endpoints, randomization procedure, blinding, sample size estimation, and statistical analysis plan. There are only several minor trial design and statistical concerns.

Statistical critiques:

1. It is unclear why the authors chose the proportion of subjects (binary outcome) reporting the same moderate to severe symptom (among dizziness, headache, or nausea) occurring on more than one day within any given two-week recall period (after week 0–2 acclimation period) to determine the study power instead of the beverage tolerability questionnaire (BTQ) score (continuous outcome).

2. It is unclear why the authors proposed having an independent study medical officer to review the safety lab results and approve continuation of subjects in the study instead of forming a DSMB.

3. Please clearly define the “primary outcome” in the Sample Size section. In addition, the authors should focus on a precision analysis instead of a formal power analysis for their pilot study with only 30 participants.

4. The authors proposed using block allocation sequence for the randomization process. Please clearly specify the block size.

5. In addition to the formal hypothesis test, the authors should focus on the estimation of the study outcomes in the Statistical Analysis section because this is a pilot study.

6. Please clearly specify the statistical methods that will be used for generating the 95% confidence intervals for all the study outcomes.

7. The current length of the paper seems to be too long. The authors may need to move some content to the Supplementary section.

Reviewer #2: 1. Do not use 'parallel group'

2. is this pilot?

3. Do not mention pilot

4. Headline 'Study design' is okay

5. Overall okay.

Reviewer #3: This is a timely manuscript by Stubbs and Newman and colleagues that provides an overview of a randomized, placebo-controlled, clinical trial to examine the effects of a ketone ester on age-related health outcomes in older adults. There is expanding interest in geroscience, but several hurried/poorly designed studies that over-interpret data (e.g., from non-placebo-controlled studies that were primarily planned for safety) are being conducted and/or recently published. Here, the authors provide the rationale and a thoughtful and well-written description of their planned RCT.

Strengths include the study rationale, emphasis on tolerability and safety for a short-term study, and, importantly, an assessment of blood ketone concentrations/kinetics. The collection of outcomes across multiple physiological systems a-la-geroscience is also viewed favorably, although, as the authors note, they are not likely to change in this short-term study.

A few minor comments for the authors to consider:

1. A respectful suggestion: To improve readability and expand readership, the background of the abstract is a dense on lingo that will be foreign to much of the readership (e.g., geroscience interventions, inflammo-metabolic aging mechanisms, geroscience mechanisms). The introduction is outstanding, and simple phrases used there would better set the stage in the abstract. Simpler is better.

2. Many discourage descriptions such as “generalizable” or “representative” to describe clinical trial participants. Often notable differences between those that pursue research study participation and those that do not.

3. A better description of randomization would be valuable. Are their blocks planned to balance groups on BMI, comorbid conditions, or other parameters? This may ultimately enhance interpretation of data.

4. The rationale for n = 30 isn’t clear, “Based on studies of symptoms in older adults (43, 44), we predict the primary outcome rate in the placebo group will be approximately 10%. N = 30 (15 per arm) provides 36% power to detect a 25% increase (from 10% to 35%) in the proportion of subjects meeting this primary outcome in the KE condition with two-sided α=0.10.” This would seem to relate to tolerability, so perhaps better to spell out. Moreover, not sure why 36% power is stated favorably. If just a practical sample size, better to state that up front and then provide the power this provides.

5. Just a comment-- it would be ideal to capture muscle performance and physical function measures on two occasions at baseline to account for learning effects. The placebo arm diminishes this concern, but this is somewhat standard of practice.

6. Defer to the editors, but for transparency, it seems the conflicts of Stubbs and Newman should be noted in the main text.

7. Figure 2 is not legible…at all. Fully appreciate this a conversion issue, but it should be reviewed.

7. PLOS authors have the option to publish the peer review history of their article (what does this mean?). If published, this will include your full peer review and any attached files.

Reviewer #1: No

Reviewer #2: No

Reviewer #3: No

---

## [Author Response · Author response to Decision Letter 0]

22 Jan 2024

Please see attached document for full responses to reviewers comments

Reviewer 3 noted that Figure 2 was not legible. We tried several times to upload a version of figure 2 that was not corrupt/blurred in the proofs. The files we uploaded were not corrupt when dowloaded from the 'upload documents' page but were always corrupt/blurred in the proofs. If there is a problem with the file, we can share it with the journal using some other method.

---

## [Decision Letter · Decision Letter 1]

16 Jul 2024

Rationale and protocol for a safety, tolerability and feasibility randomized, parallel group, double-blind, placebo-controlled, pilot study of a novel ketone ester targeting frailty via immunometabolic geroscience mechanisms.

PONE-D-23-22898R1

Dear Dr. Stubbs,

We’re pleased to inform you that your manuscript has been judged scientifically suitable for publication and will be formally accepted for publication once it meets all outstanding technical requirements.

Kind regards,

Ian James Martins, PhD

Academic Editor

PLOS ONE

Additional Editor Comments (optional):

Reviewers' comments:

Reviewer's Responses to Questions

**Comments to the Author**

1. Does the manuscript provide a valid rationale for the proposed study, with clearly identified and justified research questions?

Reviewer #1: Yes

Reviewer #3: Yes

Reviewer #4: Yes

2. Is the protocol technically sound and planned in a manner that will lead to a meaningful outcome and allow testing the stated hypotheses?

Reviewer #1: Yes

Reviewer #3: Yes

Reviewer #4: Yes

3. Is the methodology feasible and described in sufficient detail to allow the work to be replicable?

Reviewer #1: Yes

Reviewer #3: Yes

Reviewer #4: Yes

4. Have the authors described where all data underlying the findings will be made available when the study is complete?

Reviewer #1: Yes

Reviewer #3: Yes

Reviewer #4: Yes

5. Is the manuscript presented in an intelligible fashion and written in standard English?

Reviewer #1: Yes

Reviewer #3: Yes

Reviewer #4: Yes

6. Review Comments to the Author

You may also provide optional suggestions and comments to authors that they might find helpful in planning their study.

Reviewer #1: The authors have responded well to the statistical issues raised in the previous review. There is no further statistical concern about this revised manuscript.

Reviewer #3: Thoughtful response to the review and appropriate edits made to the revised manuscript. No additional concerns.

Reviewer #4: #1 The authors mention uncontrolled liver and kidney disease as exclusionary diagnoses, but please describe the specific criteria.

#2 Regarding allocation, are there any allocation criteria?

7. PLOS authors have the option to publish the peer review history of their article (what does this mean?). If published, this will include your full peer review and any attached files.

Reviewer #1: No

Reviewer #3: No

Reviewer #4: No

---

## [Editor Report · Acceptance letter]

23 Jul 2024

PONE-D-23-22898R1 

PLOS ONE

Dear Dr. Stubbs, 

I'm pleased to inform you that your manuscript has been deemed suitable for publication in PLOS ONE. Congratulations! Your manuscript is now being handed over to our production team.

Kind regards, 

on behalf of

Dr. Ian James Martins 

Academic Editor

PLOS ONE